# (−)-Gallocatechin Gallate: A Novel Chemical Marker to Distinguish *Triadica cochinchinensis* Honey

**DOI:** 10.3390/foods13121879

**Published:** 2024-06-14

**Authors:** Huizhi Jiang, Zhen Li, Shiqing Zhong, Zhijiang Zeng

**Affiliations:** 1Honeybee Research Institute, Jiangxi Agricultural University, Nanchang 330045, China; jhz19991221@163.com (H.J.); zhongsqing@yeah.net (S.Z.); 2Jiangxi Province Key Laboratory of Honeybee Biology and Beekeeping, Jiangxi Agricultural University, Nanchang 330045, China; 3College of Life Science and Resources and Environment, Yichun University, Yichun 336000, China; zhenli1995@aliyun.com

**Keywords:** *Triadica cochinchinensis*, honey, LC-MS/MS, (−)-gallocatechin gallate, chemical markers

## Abstract

*Triadica cochinchinensis* honey (TCH) is collected from the nectar of the medicinal plant *T. cochinchinensis* and is considered the most important honey variety in southern China. TCH has significant potential medicinal properties and commercial value. However, reliable markers for application in the authentication of TCH have not yet been established. Herein, a comprehensive characterization of the botanical origin and composition of TCH was conducted by determining the palynological characteristics and basic physicochemical parameters. Liquid chromatography tandem-mass spectrometry (LC-MS/MS) was used to investigate the flavonoid profile composition of TCH, *T. cochinchinensis* nectar (TCN) and 11 other common varieties of Chinese commercial honey. (−)-Gallocatechin gallate (GCG) was identified as a reliable flavonoid marker for TCH, which was uniquely shared with TCN but absent in the other 11 honey types. Furthermore, the authentication method was validated, and an accurate quantification of GCG in TCH and TCN was conducted. Overall, GCG can be applied as a characteristic marker to identify the botanical origin of TCH.

## 1. Introduction

Honey is a natural sweet substance produced by honeybees (*Apis mellifera*) and derived from the nectar or secretions of plants or the excretions of plant-sucking insects on the living parts of plants [1]. It is known that the chemical composition and biological activities of honey are mainly influenced by nectar source plants [2].

Honey contains approximately 20% water and 75% carbohydrates (mainly fructose and glucose), as well as flavor components, proteins, minerals and phenolic compounds. These trace components contribute candidate markers tracing the botanical origin of honey [3,4]. Among the minor components, phenolic compounds are often paid more attention due to their contribution to the unique chemical profile of honey and used as chemical markers for distinguishing botanical origins [5]. Commonly, phenolic compounds can be divided into phenolic acids and flavonoids. According to previous reports, flavonoids are the most important phenolic constituent in honey (accounting for more than 80%) [6].

Honey is broadly classified based on its botanical origin into monofloral and polyfloral types [7]. Monofloral honey is widely considered more valuable than polyfloral honey owing to its distinct flavor and pharmacological properties and, in recent years, has witnessed increased consumer demand [8,9]. It has been suggested that several medicinal properties of plants can be carried on to monofloral honey [10], and recent studies have focused on identifying chemical markers that would enable the authentication of high-quality monofloral honey originating from medicinal plants. For instance, Manuka honey is renowned for its antibacterial properties, which can be identified by chemical markers such as methyl syringate 4-O-β-D-gentiobioside and lepteridine [11,12]. Moreover, kaempferitrin is a unique flavonoid that can be used as a marker to authenticate honey obtained from the nectar of the medicinal plant *Camellia oleifera* [13]. Furthermore, safflomin A is a novel chemical marker used for the authentication of honey derived from *Carthamus tinctorius* L. (Safflower), a well-known medicinal plant belonging to the Asteraceae family [14]. In addition, kaempferol-3-O-galactoside has been proposed as a marker for authenticating honey from *Lespedeza bicolor* Turcz., which has highly valuable and relatively rare medicinal properties. Moreover, calycosin and formononetin have emerged as markers for honey from *Astragalus membranaceus* var. *mongholicus* Hsiao [15].

*Triadica cochinchinensis* is a tree or shrub belonging to the Euphorbiaceae family (Figure 1A) and one of the main nectar plants in southern China during summer time, showing a long flowering stage and high nectar production [16,17]. Traditionally used in Chinese herbal medicine, its leaves, bark and roots can be used to treat various internal pathological conditions, including nephritis, oedema, ascites, constipation and dysuria, as well as external pathologies, such as allergic dermatitis, mastitis, bruises, wounds and snakebites [18,19]. Recent studies have shown that *T. cochinchinensis* leaf and stem parts contain various bioactive compounds, including diterpenoids, phenolics, flavonoids and tannins, which are the main components imparting antioxidant, anti-inflammatory, antibacterial, hepatoprotective and antidiabetic effects [20,21].

*T. cochinchinensis* honey (TCH) easily crystallizes, and crystallized TCH assumes a white color (Figure 1B). Previous studies have shown that TCH has demonstrated the ability of alleviating alcoholic liver damage and scavenging free radicals [21,22]. It has been described that TCH is rich in phenolic acids and flavonoids such as ellagic acid, gallic acid, naringenin and rutin [22,23]. Thus, TCH holds promise for the development of dietary supplements and medicinal agents.

Being a major commercial honey in southern China, *T. cochinchinensis* nectar (TCN) production is substantial and stable, making it one of the key monofloral honeys with significant economic benefits for beekeepers [24,25]. Therefore, identifying characteristic chemical markers in TCH derived from medicinal plants is crucial. Such an identification process might facilitate exploring efficacious and potential applications of TCH, thus contributing to further popularizing, while also enhancing, both the health benefits and commercial value of this unique monofloral honey. In addition, the identification of chemical markers would contribute in the evaluation of honey authenticity, which would help to promote consumer trust and sustain the reliable development of the food industry [26].

Previous studies have shown that chromatography and mass spectrometry are essential in identifying chemical markers specific to monofloral honey [27,28]. In particular, plant-nectar-derived flavonoids found in honey are additionally considered bioactive compounds and might be used as biomarkers for discriminating honey botanical origin as well as adulteration [29]. Several studies have proposed the use of flavonoids as unique chemical markers for specific monofloral honey [13,30]. Thus, utilizing distinct flavonoid markers might be considered a reliable strategy for authenticating TCH.

The present study aimed to identify distinctive flavonoid markers in TCH using targeted metabolomics. A palynological analysis was applied to confirm the botanical origin of TCH, and basic physicochemical parameters of TCH were analyzed. Subsequently, liquid chromatography tandem-mass spectrometry (LC-MS/MS) was used to identify flavonoid types in TCH, and a comparative analysis was conducted against 11 other common types of Chinese commercial honey to discriminate unique flavonoid markers exclusive to TCH. Finally, a method for TCH authentication and accurate quantification of flavonoid markers in TCH and TCN was proposed. This study will be helpful for authenticity assessment and quality control of TCH products.

## 2. Materials and Methods

### 2.1. Chemicals and Reagents

LC-MS-grade acetonitrile and methanol were purchased from Merck (Darmstadt, Germany). LC-MS-grade formic acid was purchased from Sigma-Aldrich (St. Louis, MO, USA). Ultra-pure water was obtained from a Milli-Q Plus system (Millipore, Bradford, PA, USA). All remaining standards (purity > 98%) were purchased from MedChemExpress (Shanghai, China). Detailed information on standards used in the present study is listed in Appendix A.

### 2.2. Sample Collection

We selected the Weimin honeybee apiary in Yongxiu County, Jiujiang City, Jiangxi Province (29°01′ N, 115°29′ E) with *T. cochinchinensis* plantation areas to produce and harvest. TCH was collected from 28 May to 10 July 2023 when the flowers were blooming. TCN was collected from *T. cochinchinensis* plants by using a micro aspirator (Beijing Dalong Company Limited, Beijing, China).

Eleven types of popular commercial honey in China were selected as comparison honey, including nine types of monofloral honey, namely, *Brassica napus* honey (BNH), *Citrus reticulata* honey (CRH), *Robinia pseudoacacia* honey (RPH), *Ziziphus jujuba* honey (ZJH), *Vitex negundo* honey (VNH), *Litchi chinensis* honey (LCH), *Lycium chinense* honey (YCH), *Eriobotrya japonica* honey (EJH), *Tilia tuan* honey (TTH) and one polyfloral honey (POH) which were obtained from Wuhan Baochun Bee Products Company (Wuhan, China). *C. oleifera* honey (COH) was provided by Lishui Lantian Apiary in Changning City, China. All samples were prepared as three biological replicates and stored at −20 °C until analysis.

### 2.3. Sample Preparation

TCH and TCN samples were freeze-dried and ground into powder using a ball mill (MM400, Retsch, Haan, Germany) (30 Hz, 1.5 min). Then, 0.20 ± 0.01 g of samples was accurately weighted and mixed with 5 mL of 70% methanol and 100 μL of the internal standard working solution (daidzein, rutin and (−)-gallocatechin) of 4000 nmol/L. After ultrasonication for 30 min, samples were centrifuged at 12,000 r/min (the corresponding g-force was 11,304× *g*) for 5 min at 4 °C. The obtained supernatant was harvested and filtered through a 0.22 μm filter membrane into a glass vial for subsequent LC-MS/MS analysis.

### 2.4. Palynological Identification and Physicochemical Analysis

Palynological analysis was conducted with a light microscope equipped with a camera (DS-Fi3, Nikon Corporation, Tokyo, Japan). Honey and pollen grain samples were prepared based on the method of Song et al. [31,32]. The concentration of *T. cochinchinensis* plant pollen grains in TCH pollen grains was determined. The morphology, length and width of *T. cochinchinensis* plants and TCH pollen grains were measured. The native pollen rate and morphological characterization of the other 10 types of monofloral honey samples were also determined with reference to the previous studies [33]. The contents of fructose, glucose, sucrose, water, 5-hydroxymethylfurfural (5-HMF), as well as diastase activity, electrical conductivity, ash content and color value in TCH samples were determined in accordance with the AOAC official method [34]. Additionally, free acidity in TCH was determined based on the equivalence point titration following the method of Li [13]. The concentration of pollen grains was calculated using a hemocytometer, and the counting method used for honey samples and pollen grains was conducted as proposed by Song [31]. Minerals Fe, Cu and Zn were identified using inductively coupled plasma optical emission spectrometry (ICP-MS 7500, Agilent Technologies Inc., Santa Clara, CA, USA) [35].

### 2.5. LC-MS/MS Analysis

For quantitative and qualitative analysis of flavonoid compounds in honey and nectar samples, an ultra-performance liquid chromatography system (ExionLC™ AD, SCIEX, Framingham, MA, USA) coupled with tandem mass spectrometry (QTRAP^®^ 6500+, SCIEX, Framingham, MA, USA) was used. UPLC conditions were as follows: Waters ACQUITY UPLC HSS T3 C18 column (1.8 µm, 100 mm × 2.1 mm, Waters, Milford, MA, USA); mobile phase A, ultrapure water with 0.05% (*v*/*v*) formic acid; mobile phase B, acetonitrile with 0.05% (*v*/*v*) formic acid; flow rate, 0.35 mL/min; column temperature, 40 °C; injection volume, 2 μL. Gradient elution program was as follows: 0.0~1.0 min, 10~20% B; 1.0~9.0 min, 20~70% B; 9.0~12.5 min, 70~95% B; 12.5~13.5 min, 95%B; 13.5~13.6 min, 95~10% B; 13.6~15 min, 10% B.

Mass spectrometry (MS) conditions were as follows: electrospray ionization (ESI) source temperature, 550 °C; voltage in positive ion mode, 5500 V; voltage in negative ion mode, −4500 V; curtain gas (nitrogen), 35 psi. The collision gas was nitrogen. Each ion pair was scanned for detection in a Q-Trap 6500+ system (SCIEX, Framingham, MA, USA) on the basis of the optimized declustering potential (DP) and collision energy (CE). The specific flavonoid standards monitored in positive and negative ion modes are listed in Appendix A.

### 2.6. Construction of Flavonoid Standard Curves and Determination of Linearity Range

All 204 flavonoid standards were prepared into master batches of 10 mmol/L by methanol/water (70:30), then diluted in methanol/water (70:30) into standard curve working solutions at 0.5 nmol/L, 1 nmol/L, 5 nmol/L,10 nmol/L, 20 nmol/L, 50 nmol/L, 100 nmol/L, 200 nmol/L, 500 nmol/L, 1000 nmol/L, 2000 nmol/L. In addition, 100 µL of the internal standard working solution (daidzein, rutin and (−)-gallocatechin) of 4000 nmol/L had to be added to each working solution, and the final volume of each working solution was 5 mL. With the concentration ratio of the external standard to the internal standard as the horizontal coordinate and the area ratio of the external standard to the internal standard ratio as the vertical coordinate, standard curves were constructed from the mass spectral peak intensity data of the corresponding quantitative signals of each standard working solution. A good linearity of 204 flavonoid standard curves was determined within the concentration range of 0.5 nmol/L to 200 nmol/L (R2 ≥ 0.9900), and the results are shown in Appendix A.

### 2.7. Determination of Qualitative and Quantitative Parameters of the LC-MS/MS Method

A database was constructed based on flavonoid standards, and MS data were analyzed qualitatively. The multiple reaction monitoring (MRM) mode of triple quadrupole MS was used for quantitative analysis, in which precursor ions of the target substance (parent ions) were initially screened by the quadrupole, and ions corresponding to substances with other molecular weights were excluded to limit interference. Precursor ions were then induced to ionization by the collision chamber, broken to form multiple fragment ions and filtered by the triple quadrupole for the selection of fragment ions with the required characteristics, and the interference of non-target ions was simultaneously excluded, resulting in more accurate and reproducible quantification. MS data of honey and nectar samples were obtained, and chromatographic peaks of all targets were integrated and quantitatively analyzed based on the flavonoid standard curves.

### 2.8. Data Processing

Palynological and physicochemical analysis had six replicates, and flavonoid analysis had three replicates. Mass spectrometry data acquisition was conducted in Analyst 1.6.3 software (AB SCIEX, MA, Framingham, USA). Multiquant 3.0.3 software (AB SCIEX, Framingham, MA, USA) was used for mass spectrometry data procession, and the accuracy of metabolite quantification was referenced to the retention time and peak shape information of the standards and mass spectrometry peaks of the analytes after integral correction. The results are expressed as mean ± standard deviation (SD).

## 3. Results and Discussion

### 3.1. Palynological and Physicochemical Characterization of Honey

Firstly, approximately 89.60 ± 2.60% of pollen grains in TCH matched those collected directly from *T. cochinchinensis* plants. Pollen grains from TCH exhibited a prolate shape in the equatorial view (Figure 1C) and a trilobed circular shape in the polar view (Figure 1D), measuring 44 × 22 μm in size with tricolporate and reticulate pattern on the outer wall, thus surpassing the 45% requirement to be considered a monofloral honey [36].

Palynological analysis found that the other 10 types of monofloral honey samples also had a high single pollen rate (Table 1), and the specific pollen morphology is shown in Figure 2.

Table 2 depicts the main physicochemical parameters of TCH. Honey is mostly composed of sugars [37]. The content of total reducing sugars was 74.74% (fructose content, 37.42%; glucose content, 37.32%), and sucrose content was 0.77%, all of which are in accordance with European Union honey standards [38]. The fructose/glucose (F/G) ratio in honey is an important parameter in predicting the crystallization of honey, and when the ratio is <1.11, honey will crystallize very easily [39]. TCH had an (F/G) ratio of 1.00, indicating fast crystallization.

Water content is one of the significant parameters used to identify the quality of honey and is taken as a vital indicator for maturity, viscosity and stability [40]. The water content of TCH was 18.65%, which matched the standards that the water content should not exceed 20% [38].

Moreover, 5-HMF is a cyclic aldehyde, an intermediate product from the Maillard reaction (a non-enzymatic browning reaction) during processing or long storage of honey. 5-HMF content is widely recognized as a parameter affecting honey freshness [41]. The content of 5-HMF was 1.87 mg/kg, which is within the acceptable range for honey samples based on European Union [38].

Diastases play important roles in the process of honey maturation, whose function is to digest the starch molecule in a mixture of maltose and maltotriose [42,43]. Diastase content depends on the different floral and geographical origins of the honey. TCH exhibited a diastase activity of 2.55 mL/ (g h), which is consistent with the findings of Liu et al. who described naturally low diastase activity in TCH [22].

Trace minerals are important constituents of honey and play specific roles in human health [44]. The presence of Fe in honey can alleviate anaemia and increase immunity in honey eaters [45]. Cu is necessary for normal human health and growth and contributes to immune function [46]. Zn is an essential antioxidant mineral that can promote wound healing and decrease risks of cancer and cardiovascular diseases [47]. The mineral concentrations of Fe, Cu and Zn in TCH were 6.23 mg/kg, 105.31 μg/kg and 5.41 mg/kg, respectively. According to a previous study conducted by our research group, TCH exhibited high contents of Fe and Zn compared to other nine types of honey [48].

Taken together, these findings indicate that TCH can be considered a high-quality honey with a unique flavor.

### 3.2. Screening and Identification of Unique Flavonoid Markers in TCH

The class of flavonoids comprises over 50% of phenolic compounds, which are the essential products of secondary plant metabolism [43]. The flavonoid composition of honey is mainly associated with its floral source and geographical origin [49], thus serving as a tool for honey classification and authentication, particularly for monofloral honeys [50].

A total of 35 flavonoids were detected in TCH using LC-MS/MS (Figure 3A). Compared to the other 11 types of honey tested herein, TCH was found to contain 12 distinctive flavonoids, which was also the largest number among all tested honeys. These 12 flavonoids included 5,7,3′,4′-tetramethoxyflavone, genistin, neohesperidin, mangiferin, epigallocatechin, oroxinA, catechingallate, sieboldin, (−)-gallocatechin gallate, dihydromyricetin, myricitrin and naringenin-7-glucoside (Figure 3B).

In addition, 18 flavonoids were shared between TCH and TCN (Figure 3C). A comparative analysis between the 18 flavonoids and the 12 flavonoids previously found to be unique to TCH among the 12 types of honey revealed that only one flavonoid, (−)-gallocatechin gallate (GCG), was uniquely found in both TCH and TCN (Figure 3D). Therefore, GCG was considered a distinct flavonoid marker of TCH. Chromatographic and MS spectra of GCG in TCH and TCN, as well as of the GCG standard, are shown in Figure 4.

### 3.3. Method Validation and Quantification of (−)-Gallocatechin Gallate

GCG, a catechin compound, has various health benefits, including antioxidant and antibacterial properties [51], cholesterol- and triglyceride-lowering effects [52], melanin synthesis inhibition [53] and neuroprotective and cardioprotective effects [54,55]. Thus, detecting the accurate quantitative measurement of GCG in TCH would allow evaluating its authenticity and also contribute to popularizing this characteristic monofloral honey.

Herein, an LC-MS/MS method was developed to quantify GCG in TCH and TCN. To achieve this, 11 GCG standard solutions were used to construct a standard curve, allowing the detection of GCG content in TCH and TCN samples. The GCG standard curve was described by the equation y = 1010.75160x − 2834.40065 (R^2^ ˃ 0.99) within the linear range of 5–2000 nmol/L. The limits of detection (LOD), estimated to a signal to noise (S/N) ratio of 3, and the limits of quantification (LOQ), estimated to a signal to noise (S/N) ratio of 10, for GCG were 1.14 nmol/kg and 3.43 nmol/kg, respectively. Based on the standard curve, the content of GCG in TCH and TCH was 130.78 ± 4.44 nmol/kg and 96.33 ± 2.16 nmol/kg, respectively. Moreover, the relative standard deviation (RSD) of TCH and TCN was 3.40% and 2.25%, respectively. The combined results are shown in Table 3. Taken together, the LC-MS/MS method developed herein could be considered sensitive and reliable for the detection of GCG in TCH and TCN.

## 4. Conclusions

In this study, the LC-MS/MS method was applied to identify flavonoids in TCH, TCN and 11 other types of commonly commercial Chinese honey to accurately identify and quantify the characteristic markers of TCH. GCG was identified as a unique flavonoid marker for TCH. In addition, a reliable and accurate LC-MS/MS method was established for the first time to identify GCG in TCH and TCN. Thus, the findings of the present study provide a novel and reliable solution for the authentication and quality control of TCH, which would also provide theoretical support for developing standards for TCH products.

## Figures and Tables

**Figure 1 foods-13-01879-f001:**
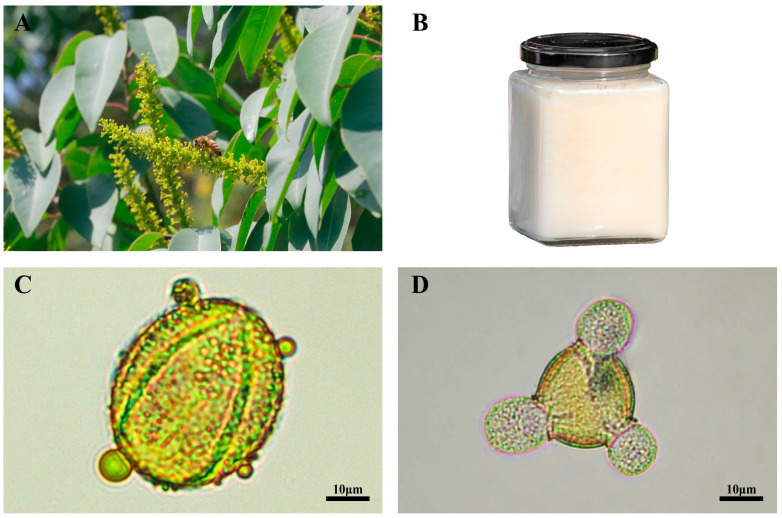
The botanical origin and palynological analysis of TCH. (**A**) Honeybee-visited *T. cochinchinensis* flower. (**B**) Crystallized *T. cochinchinensis* monofloral honey. (**C**) An equatorial view of TCH pollen grains under the light microscope. (**D**) A polar view of TCH pollen grains under the light microscope.

**Figure 2 foods-13-01879-f002:**
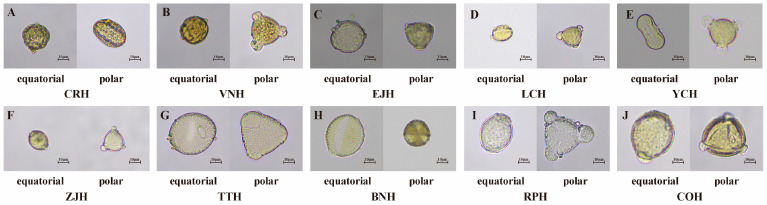
The pollen morphology of 10 common types of Chinese commercial monofloral honey. (**A**): *Citrus reticulata* honey (CRH); (**B**): *Vitex negundo* honey (VNH); (**C**): *Eriobotrya japonica* honey (EJH); (**D**): *Litchi chinensis* honey (LCH); (**E**): *Lycium chinense* honey (YCH); (**F**): *Ziziphus jujuba* honey (ZJH); (**G**): *Tilia tuan* honey (TTH); (**H**): *Brassica napus* honey (BNH); (**I**): *Robinia pseudoacacia* honey (RPH); (**J**): *Camellia oleifera* honey (COH).

**Figure 3 foods-13-01879-f003:**
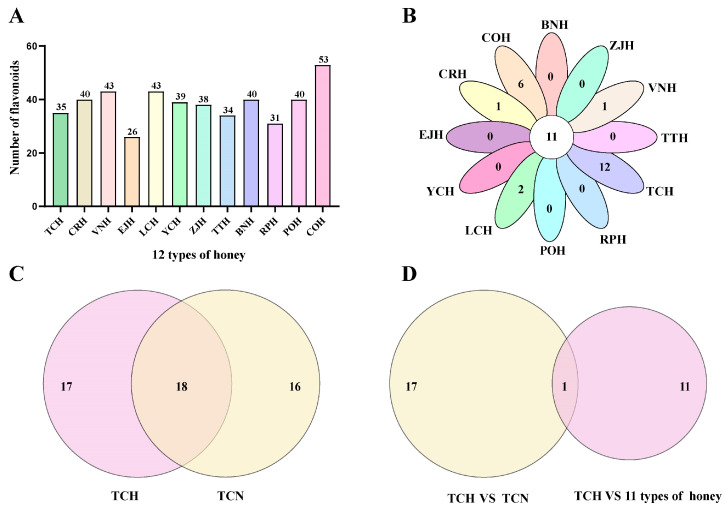
(**A**) Flavonoid species identified in TCH and 11 other common types of Chinese commercial honey. (**B**) Flavonoid species distinctive in one type of honey relative to 11 other common types of Chinese commercial honey. TCH has 12 distinctive flavornoid species. (**C**) The flavonoid species that are shared in TCH and TCN. (**D**) Flavonoids distinctive to TCH and TCN relative to 11 other common types of Chinese commercial honey.

**Figure 4 foods-13-01879-f004:**
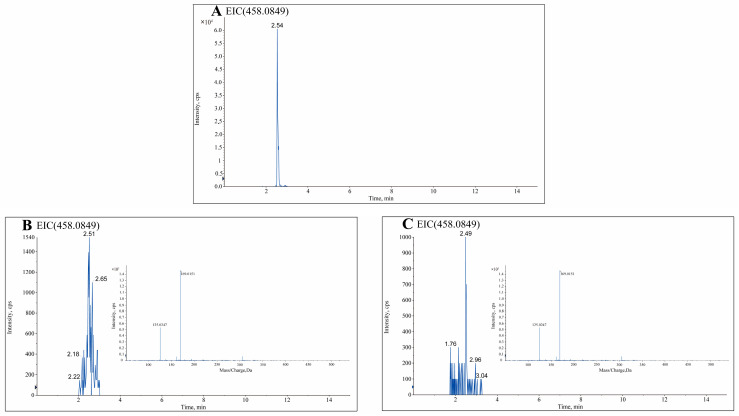
The typical chromatograms of the extracts from the positive/negative mode analyzed by LC-MS/MS. (**A**) Extracted ion chromatogram (EIC) of GCG standard. (**B**) EIC and mass spectrum of GCG in TCN. (**C**) EIC and mass spectrum of GCG in TCH.

**Table 1 foods-13-01879-t001:** Native pollen rate of 10 common types of Chinese commercial monofloral honey.

Honey Variety	Native Pollen Rate (%)	Honey Variety	Native Pollen Rate (%)
*Citrus reticulata* honey (CRH)	75.94 ± 5.22	*Ziziphus jujuba* honey (ZJH)	86.41 ± 2.22
*Vitex negundo* honey (VNH)	64.81 ± 5.50	*Tilia tuan* honey (TTH)	88.80 ± 1.25
*Eriobotrya japonica* honey (EJH)	76.82 ± 6.62	*Brassica napus* honey (BNH)	84.85 ± 2.08
*Litchi chinensis* honey (LCH)	85.51 ± 2.09	*Robinia pseudoacacia* honey (RPH)	71.47 ± 6.49
*Lycium chinense* honey (YCH)	81.08 ± 2.46	*Camellia oleifera* honey (COH)	87.76 ± 6.63

**Table 2 foods-13-01879-t002:** Physicochemical parameters of TCH.

Parameter	Mean ± SD	Units
Fructose	37.42 ± 0.71	%
Glucose	37.32 ± 0.36	%
Sucrose	0.77 ± 0.10	%
Water	18.65 ± 0.56	%
HMF	1.87 ± 0.12	mg/kg
Diastase activity	2.55 ±0.32	mL/(g·h)
Electrical conductivity	0.14 ± 0.003	mS/cm
ash content	0.07 ±0.001	g/100 g
Color value	32.00 ± 0.00	mm Pfund
Free acidity	11.82 ± 0.22	mL/kg
Pollen grains concentration	18,025.00 ± 641.67	grain/mL
Fe	6.23 ± 0.05	mg/kg
Cu	105.31 ± 5.98	μg/kg
Zn	5.41 ± 0.29	mg/kg

**Table 3 foods-13-01879-t003:** GCG of standard curve, LOD, LOQ and the content of GCG in TCH and TCN.

Compound	Standard Curve	LOD (nmol/kg)	LOQ (nmol/kg)	Regression (R^2^)	TCH (*n* = 3)		TCN (*n* = 3)	
					Content (nmol/kg)	RSD (%)	Content (nmol/kg)	RSD (%)
GCG	y = 1010.75160x − 2834.40065	1.14	3.43	0.9994	130.78 ± 4.44	3.40	96.33 ± 2.16	2.25

## Data Availability

The original contributions presented in the study are included in the article/Appendix A, further inquiries can be directed to the corresponding author.

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
