# Peer review of "(−)-Gallocatechin Gallate: A Novel Chemical Marker to Distinguish *Triadica cochinchinensis* Honey"

_foods, 2024, doi:10.3390/foods13121879_

Round 1
Reviewer 1 Report
Comments and Suggestions for Authors
With the proposed publication, the authors intend to identify a reliable flavonoid marker fot Triadica cochinchinesis honey (TCH) delivered from its nectar (TCN).
The introduction seems complete and well written focusing on which important topics to be addressed.
The methodology used is well conducted and implies the answers needed.
However, the use of 204 standards seems excessive; using far fewer standards, it would be possible to identify the compounds through their mass-to-charge ratio and retention time.
Section 2.4 - It is not clear to me why a physicochemical analysis in particular determination of minerals by ICP-MS it is needed to reach the answer asked in this manuscript. Is it important for the identification of the honey variety?
Section 2.8 it is not needed. That calculations are standard calculation for the calculation of bioactive compounds from the peak area.
The conclusions need improvement. Some considerations on the possibility of incorporating these results into laws and standards could be interesting.
Reviewer 2 Report
Comments and Suggestions for Authors
Manuscript ID: foods-3025546
Type of manuscript: Article
Title: (−)-Gallocatechin Gallate: A Novel Chemical Marker to Distinguish Triadica cochinchinensis Honey
The article is written in a very good English and totally fulfills the requirements for good scientific work.
Hui Zhi Jiang and his co-workers wrote a manuscript on the flavonoid content of Triadica cochinchinensis Lour. honey (TCH) and nectar (TCN) as well as eleven other common commercial honey varieties. The aim was to find unique markers to distinguish TCH and TCN from other types of honey.
In order to do this, the authors carried out a targeted metabolomic study investigating 204 different flavonoids by UPLC-MS/MS.
Why the authors have not done a untargeted study using parent ion scan or or even neutral loss scan to find unknown flavonoids characteristic for TCH and TCN?
The authors could show that gallocatechin gallate (GCG) is a flavonoid found only in the analysed TCH and TCN samples and not in the analysed samples of different honey varieties. GCG was the only substance unique for TCH and TCN.
What about unique markers for the other honey varieties? The analysis of GCG in TCH can not detect "contaminations" or admixtures of cheaper honey to TCH.
The authors did not demonstrate the detection limit of adulteration of different honeys with TCH.
The authors did not analyse TCH from different geographical origins or different years in order to demonstrate that GCG is a stable and reliable marker for TCH.
In addition to flavonoids, typical honey parameters such as sugars, HMF, acidity, colour and diastase activity were determined. The content of Fe, Cu, and Zn was also determined. The authors should explain why these three minerals were determined and only these three.
some small remarks:
line 96: produce
line 101ff: Please always or never use the author quote
line 109: which device was used for grinding
line 110: 0.20 ± 0.01 g
line 112: please add rotor diameter or g-force
line 141ff: which curtain gas, which collision gas
line 221ff: mS/cm not ms/cm
µg/kg not ug/kg
page 9: figure 4 is of poor technical quality
line 256: please compare the content of GCG in honey with it biological activity! What quantity of honey is required to sustain this level of activity?
I highly recommend a few very minor changes.
